# Epidermal Growth Factor Receptor Inhibitors in Glioblastoma: Current Status and Future Possibilities

**DOI:** 10.3390/ijms25042316

**Published:** 2024-02-15

**Authors:** Shawyon Ezzati, Samuel Salib, Meenakshisundaram Balasubramaniam, Orwa Aboud

**Affiliations:** 1California Northstate University College of Medicine, Elk Grove, CA 95757, USA; shawyon.ezzati9555@cnsu.edu (S.E.); samuel.salib10011@cnsu.edu (S.S.); 2Reynolds Institute on Aging, Department of Geriatrics, University of Arkansas for Medical Sciences, Little Rock, AR 72205, USA; mbalasubramaniam@uams.edu; 3Department of Neurology, Department of Neurological Surgery, Comprehensive Cancer Center, University of California, Davis, Sacramento, CA 95817, USA

**Keywords:** glioblastoma, epidermal growth factor receptor, tyrosine kinase, tyrosine kinase inhibitors, precision oncology, molecular heterogeneity, biological therapy

## Abstract

Glioblastoma, a grade 4 glioma as per the World Health Organization, poses a challenge in adult primary brain tumor management despite advanced surgical techniques and multimodal therapies. This review delves into the potential of targeting epidermal growth factor receptor (EGFR) with small-molecule inhibitors and antibodies as a treatment strategy. EGFR, a mutationally active receptor tyrosine kinase in over 50% of glioblastoma cases, features variants like EGFRvIII, EGFRvII and missense mutations, necessitating a deep understanding of their structures and signaling pathways. Although EGFR inhibitors have demonstrated efficacy in other cancers, their application in glioblastoma is hindered by blood–brain barrier penetration and intrinsic resistance. The evolving realm of nanodrugs and convection-enhanced delivery offers promise in ensuring precise drug delivery to the brain. Critical to success is the identification of glioblastoma patient populations that benefit from EGFR inhibitors. Tools like radiolabeled anti-EGFR antibody 806i facilitate the visualization of EGFR conformations, aiding in tailored treatment selection. Recognizing the synergistic potential of combination therapies with downstream targets like mTOR, PI3k, and HDACs is pivotal for enhancing EGFR inhibitor efficacy. In conclusion, the era of precision oncology holds promise for targeting EGFR in glioblastoma, contingent on tailored treatments, effective blood–brain barrier navigation, and the exploration of synergistic therapies.

## 1. Introduction

Glioblastoma is the most common and aggressive primary brain tumor in adults. The World Health Organization (WHO) classifies glioblastomas as grade 4 gliomas composed of pleomorphic tumor cells with poor astrocytic differentiation and a high mitotic rate. High-grade astrocytomas are also classified by their state of isocitrate dehydrogenase (IDH) as either wildtype, or mutant, with the term glioblastoma reserved to IDH wildtype astrocytoma WHO grade 4 [1]. Patients with mutated IDH have better overall survival (OS). Another biomarker important in the prognostication of glioblastoma is O^6^-methylguanine-DNA methyltransferase (MGMT) methylation. Patients with MGMT methylation also have better OS [2]. These markers aid in treatment path and provide insight to patient prognosis upon diagnosis. Although the WHO classification of glioblastoma is straightforward, it is a diverse disease because of its intra- and inter-tumor heterogeneity and invasive growth, making treatment anything but straightforward [3].

Patients with glioblastoma have a median survival of 15 months despite many treatment approaches [4]. Surgical resection is the first step in treatment. Improvements on glioblastoma surgical resection have been made because of its positive impact on survival. The addition of neuromonitoring, Laser Interstitial Thermal Therapy (LITT), fluorescence-guided surgery, and mass spectrometry have improved the precision of surgery. The current standard of care treatment for glioblastoma includes maximal safe surgical resection, followed by radiation therapy and adjuvant chemotherapy [1,2]. These treatments have been shown to improve survival and quality of life but leave a high recurrence rate [3,5]. Surgical techniques have been improved, but the development of more specific and effective drugs is needed to control glioblastoma.

Neuro-oncology is one of the fields seeking to benefit from a revolutionary molecular era. Advancements in histology and immunochemistry are allowing for increased precision in diagnosis, treatment, and prognosis. The WHO’s CNS tumor nomenclature and grading are frequently updated as knowledge of tumor molecular biology increases. Updates have been made in taxonomy based on increased understanding of biomarkers, molecular structures, and genetics. Grading has been modernized by using the presence of certain molecular markers, rather than relying on histology [6]. Incorporating molecular biomarkers into tumor diagnosis allows for neuro-oncologists to configure treatments and factor patient prognosis into treatment options.

One potential target for therapy is epidermal growth factor receptor (EGFR). EGFR is an oncogenic receptor tyrosine kinase (RTK) that plays a crucial role in cell growth, survival, and proliferation by a downstream regulation of mitogen-activated protein kinase (MAPK) and phosphatidylinositol 3 (PI3K) kinase growth pathways [7]. In glioblastoma, EGFR displays the highest enrichment of any gene. It is mutationally active in over 50% of tumors, contributing to tumor growth, invasion, angiogenesis, and therapeutic resistance [8,9,10]. Its prevalence makes it a promising pharmacological target for small-molecule inhibitors and antibodies.

## 2. EGFR Receptors: Structure and Signaling Pathways

EGFRs are 1186 amino acid transmembrane receptors that can be activated by a variety of ligands binding its extracellular domain (ECD) [3]. EGFRs are part of the ErbB family, including EGFR (Erb1/HER1), HER2 (ErbB2), HER3 (Erb3), and HER4 (ErbB4) [7]. Each of these RTKs have different ligand-binding specificities, including epidermal growth factor (EGF), transforming growth factor (TGF- α), heparin binding-EGF (HB-EGF), betacellulin, amphiregulin, epiregulin, and epigen [11]. Activation leads to homo- or hetero-dimerization of the receptors with members of the ErbB family, and cytoplasmic autophosphorylation of downstream MAPK, PI3K, and signal transducer and activator of transcription 3 (STAT3) pathways. The EGFR receptor pathways converge onto the mammalian target of the rapamycin (mTOR) pathway and play an essential role in its regulation. These pathways regulate cell proliferation, survival, migration, and angiogenesis [9,12]. When left constitutively active by oncogenic mutations, uncontrollable cell growth arises.

## 3. EGFR Mutations in Glioblastoma

Various EGFR variants have been associated with glioblastoma. Most of these mutations are of the ECD, but some involve the intracellular domain (ICD). Amongst others, the most common include EGFRvIII, EGFRvII, EGFRvIV, EGFRx, and missense mutations. These mutants can dimerize with EGFR receptors or other ErbB receptors like HER2. Their presence creates increased heterogeneity in glioblastoma tumors and to target them therapeutically, their structure must be well understood.

ECD domain mutations are the most common in glioblastoma. The EGFRvIII variant, characterized by a deletion of exons 2–7, exerts its influence on the ECD. This mutation results in the persistent activation of downstream signaling pathways and a predilection of the PI3K signaling cascade over the MAPK and STAT3 pathways, regardless of ligand presence or receptor conformation. Notably, it is the most prevalent variant found in glioblastomas and when it co-occurs with amplified EGFR, it results in a poor prognosis [8,9,13]. Similarly, the EGFRvII variant is a deletion of exons 14–15, leaving the receptor constitutively active regardless of ligand presence or conformation. The mutation impacts the ECD and leads to a downstream predilection for PI3K signaling. It is the second most prevalent variant found in glioblastoma [14,15]. Mis-sense mutations of EGFR ECD constitute about 10% of glioblastoma cases. These mutations leave the receptor constitutively active and patients with these mutations have significantly shorter survival [16,17,18,19]. EGFRvIII, EGFRvII, and mis-sense mutations account for the majority seen in patients with EGFR-driven glioblastoma. Another variant, EGFRx, a deletion of exons 2–14, leaves the receptor without a binding domain. It was found to be required for tumor proliferation and tumorigenesis in glioblastoma xenografts. This variant activates STAT5 by spontaneous asymmetric dimerization [20]. This variant signifies the importance of ECD mutations in driving glioblastoma. To successfully inhibit oncogenic EGFR in glioblastoma, it is important to target the ECD.

Intracellular mutations are less common and still play a vital role in glioblastomas’ heterogeneity. The variant EGFRvIV entails a deletion effecting the intracellular tyrosine kinase domain, resulting in persistent dimerization, and consequently, constitutive activation of STAT3 regardless of ligand presence [21]. Targeting the downstream intracellular pathways of EGFR variants, in addition to the ECD, is important to suppress glioblastoma progression.

The described variants exhibit notable similarities and convergence that make them a viable target for therapy. Firstly, most of these mutants carry ECD mutations. Glioblastoma tumors carrying these mutations have demonstrated a dependence on EGFR, and it has been shown in vitro that tumor regression occurs upon the cessation of EGFR kinase signaling [22]. Targeting these variants should effectively treat glioblastoma. Additionally, wildtype EGFR amplification is observed in 57.4% of glioblastoma cases, and it concurrently associates with each of these variants [23,24]. Finally, each of these mutations create a decreased steric hinderance of N-terminal residues of the TR1 (N-TR1) fragment within EGFR. This leaves it unstable in its constitutively untethered oncogenic state, exposing cryptic epitope 806 in each variant. Notably, amplified wildtype EGFR in glioblastoma also converges to expose this epitope [14,25,26]. The existence of this epitope makes it possible to target oncogenic variants and amplified EGFR with a single antibody. Moreover, epitope 806 emerges as a potential target for tumor therapy and identification. Targeting EGFR in glioblastoma should be aimed at glioblastoma-specific mutants, and an effective treatment should also suppress amplified wildtype EGFR.

## 4. Current EGFR Inhibitors for Cancer Therapy

EGFR is a target of therapeutics in cancers besides glioblastoma. Non-small-cell lung cancers have developed more precise treatment options based on molecular markers, like EGFR. In EGFR-driven non-small-cell lung cancer, erlotinib and gefitinib are used to target constitutively active EGFR with mutationally altered active sites [27,28]. These first-generation tyrosine receptor kinase inhibitors (TKIs) act on the intracellular tyrosine kinase domain and are effective in reducing tumor cell proliferation and survival, but their use leads to intrinsic resistance (Figure 1). This occurs when an additional gatekeeper mutation stops these drugs’ effects on EGFR by altering the kinase domain or by activating alternative growth pathways and receptors like HER2 [7,29]. Lapatinib, neratinib, and afatinib, second-generation TKIs, and osimertinib, a third-generation TKI, have been developed to overcome the latest gatekeeper mutations in non-small-cell lung cancer (Figure 1) [30,31,32]. Afatinib improved outcomes and tolerability compared to gefitinib in patients with EGFR-driven non-small-cell lung cancer [33]. Although afatinib had better results than gefitinib, these patients are still left with high rates of resistance. Improved understanding of EGFR during the progression of this disease has led to the development of osimertinib for patients who have gained resistance to all earlier generations of EGFR TKIs. Osimertinib has a higher efficacy in treating EGFR-mutant non-small-cell lung cancer than traditional platinum therapy and showed promising increases in progression-free survival [34,35,36]. There is a need for constant advancement and updates to TKIs to overcome resistance in EGFR-driven tumors.

Non-small-cell lung cancers metastasize to the brain more commonly than any other organ [37]. In the case that it does, whole-brain radiotherapy is the standard treatment. Unfortunately, this treatment only leads to a median survival of 3–6 months and does not improve the poor quality of life brought about by the disease [38]. In patients with non-small-cell lung cancer with brain metastasis, gefitinib reduced tumor size and neurological symptoms, and those that responded had rapid tumor regression [39]. A phase 2 trial using gefitinib showed a response rate of 87.8% and a median survival time of 21.9 months in patients with brain metastasis [40]. Gefitinib shows strong results in treating EGFR-driven metastatic brain cancer despite its shortcomings with resistant mutations and effective blood–brain barrier (BBB) penetration. Similar to first-generation TKIs, afatinib showed strong results in treating EGFR-driven brain metastasis of non-small-cell lung cancer. The drug showed improved progression-free survival compared to chemotherapy [41]. The continued advancement of EGFR TKIs has created reduced toxicities and improved treatment. Phase 2 trials of osimertinib showed strong results against CNS metastases. Additionally, it was found that osimertinib had increased efficacy in non-small-cell lung cancer and CNS metastasis compared to earlier EGFR TKIs [36,42]. By updating EGFR TKIs based on specific understanding of the receptor in non-small-cell lung cancer, new generations improve upon the last and lead to better treatment. The success of these drugs in brain metastasis of EGFR-driven tumors provides a roadmap for configuring their use to EGFR-driven glioblastomas.

In addition to TKIs, there are monoclonal antibodies such as cetuximab and panitumumab that target and inactivate EGFR receptors. Cetuximab is an IgG monoclonal antibody against EGFRvIII and wildtype EGFR that has been successful in the treatment of metastatic colorectal cancer and squamous head and neck cancer (Figure 1) [43,44]. Panitumumab is a monoclonal antibody antagonist of wildtype EGFR that has found success in the treatment of colorectal carcinoma [45]. Amivantamab is an EGFR antibody that works on the receptors’ extracellular domain and was found to be successful in non-small-cell lung cancer [46]. Antibodies are ideal for targeting EGFR because they are highly specific and have low toxicities. Recently, these drugs are being used in trials on patients with glioblastoma.

## 5. EGFR Inhibitors in Glioblastoma: Preclinical Studies

Preclinical studies of EGFR inhibitors in glioblastoma models have shown promising effects in reducing cell proliferation, inducing apoptosis, and inhibiting invasion and angiogenesis. EGFR dependence has been demonstrated in studies using glioblastoma cell lines with ECD EGFR mutations. Apoptosis occurred when EGFR protein expression was suppressed with retroviral shRNA [22]. Similarly, in vivo studies using glioblastoma xenografts in mice, showed slowed growth and increased apoptosis when EGFR expression was suppressed using a tetracycline-regulatable expression system [47]. Based on EGFR-driven glioblastomas’ dependence upon EGFR in vivo and in vitro, it would be assumed that TKIs should stop tumor growth.

Traditional EGFR inhibitors fail to target the ECD of the receptor and do not effectively target EGFR conformations in glioblastoma. Trials using erlotinib and gefitinib on glioblastoma tumors in vivo showed poor results in halting tumor growth or causing cell death. Lapatinib showed better results in inducing cell death. This is likely because erlotinib and gefitinib are designed to target mutations in the intracellular tyrosine kinase, whereas glioblastoma mutations are primarily in the ECD. Lapatinib has a higher affinity for the ECD, which is why it likely showed better results in vivo. Additionally, all three of these drugs target conformationally active EGFR. In glioblastoma, mutated EGFR receptors display active signaling despite inactive conformation [22]. Traditional TKIs are not designed to target EGFR in glioblastoma.

Modern immunology has made it possible to target tumor cells specifically. Compared to classic EGFR antibodies used for cancers with varying EGFR conformations, monoclonal antibody 806, targeting epitope 806, has shown promising results in human xenografts and glioblastoma tumor cells. The antibody was found to bind and internalize in xenografts overexpressing EGFR at high levels in vitro and in vivo. It also showed significant anti-tumor activity against amplified and variant EGFR [48,49,50]. Designing new biological agents that specifically target glioblastoma receptors may help minimize side effects and improve drug delivery. Monoclonal antibodies may prove to be a useful therapeutic if designed to properly target EGFR in glioblastoma.

Interestingly, EGFR inhibitors have had success when combined with other therapeutic agents in preclinical studies. Cetuximab with radiation showed effectiveness in vivo and in vitro [51]. In another study, it was found that when an EGFR inhibitor was combined with a histone deacetylase (HDAC) inhibitor, human glioblastoma cell viability and proliferation were reduced. Inhibiting HDAC and EGFR signals could be synergistically used to suppress glioblastoma [52]. Similarly, in EGFRvIII-driven glioblastomas, it was found that an anti-EGFRvIII monoclonal antibody suppressed angiogenesis and promoted apoptosis when administered with rapamycin in vivo. This is because both drugs synergistically act to reduce the activity of the PI3k pathway [53]. EGFR inhibitors are more efficient in inducing apoptosis and suppressing angiogenesis when administered with other drugs because of the synergistic inhibition of downstream growth pathways. Additionally, coadministration of drugs can create therapeutic vulnerability. Dexamethasone is used to manage inflammation in glioblastoma but has been found to have a negative impact on OS [54,55,56]. It was also found to have radio-protective properties in glioblastoma cells. Surprisingly, when TKIs were administered with dexamethasone, glioblastoma cells were more sensitive to treatment in vitro. This implies dexamethasone creates therapeutic vulnerability for TKIs [57]. In glioblastoma, coadministration of EGFR inhibitors with other drugs may increase therapeutic vulnerability of the tumor and suppression of downstream growth pathways. Overall, the results of preclinical studies suggest EGFR to be a promising glioblastoma therapeutic target, when targeted specifically and with the correct combination of drugs.

## 6. Clinical Trials with EGFR Inhibitors in Glioblastoma

Clinical trials evaluating traditional EGFR inhibitors in glioblastoma have shown variable results. Erlotinib did not have success in improving OS in patients, even when combined with temozolomide, radiation, and other synergistic drugs (Table 1) [58,59,60]. This is likely due to erlotinib not targeting the ECD of EGFR. Gefitinib was found to slightly improve median survival in patients with glioblastoma and have high concentrations in glioblastoma tissue resected from those who took the drug prior to surgery. Immunologic examination of patient tumors showed that the drug reached its target and was found to efficiently dephosphorylate EGFR; however, gefitinib alone was not enough to stop tumor growth signaling [61]. Additionally, a novel anti-EGFR antibody, GC1118, was evaluated in patients with recurrent glioblastoma. Similar to gefitinib, it was found that GC1118 effectively targeted tumor tissue and upregulated immune signatures. However, alone it was not enough to improve progression-free survival (PFS) because of tumor evolution affecting drug efficacy and an insufficient suppression of growth pathways (Table 1) [62]. These findings further suggest the need for a strong, synergistic inhibition of downstream EGFR growth pathways and the need for new TKIs that specifically target ECD variants in glioblastoma.

Like GC1118, the use of other anti-EGFR antibodies in glioblastoma has been elucidating. Depatuxizumab mafodotin is an antibody drug-conjugated form of antibody 806. A phase III trial treated newly diagnosed patients with EGFR-amplified glioblastoma with depatuxizumab mafodotin, temozolomide, and radiotherapy and found no success in improving OS. PFS improved in all patients and lasted longer in patients with the EGFRvIII variant (Table 1). These results may suggest that the antibody targets EGFRvIII more than wildtype variants. They may also suggest intrinsic resistance fueled by alternative tumor growth pathways after EGFR was suppressed by the drug [63]. Additionally, cetuximab, another antibody targeted at EGFR, showed promising responses in trials when combined with drugs like bevacizumab and irinotecan [64,65]. However, this antibody also targeted EGFR in normal tissue, causing side effects like rash and gastrointestinal toxicity. These side effects are exacerbated by large amounts of the drug needed to penetrate the BBB in order to achieve a therapeutic dose in the brain (Table 1) [66,67]. Trials using anti-EGFR antibodies have shown the need for synergistic inhibition to avoid resistance and sufficiently suppress all oncogenic growth. There is also the need for a delivery mechanism or glioblastoma EGFR-mutant-specific drug to avoid the buildup of antibodies in non-tumor tissue.

Studying clinical trials highlights EGFR’s importance as a biomarker for patients with glioblastoma. Dacomitinib, a second-generation irreversible EGFR inhibitor, which has been found to be successful in the treatment of non-small-cell lung cancer, reached phase II trials in patients with recurrent glioblastoma patients with EGFR amplification with or without EGFR variants. The drug was largely unsuccessful but had a positive effect on PFS and OS in a small number of patients (Table 1) [68]. Like other first- and second-generation TKIs, dacomitinib did not work on every patient, but it is interesting that the regimen did work in certain patients. Studying similarities and differences between populations in which this drug was successful could elucidate trends for populations that are more likely to benefit from this treatment. Similarly, when the results of cetuximab trials where studied, it was found that the drug improved OS and PFS in patients with EGFR amplification lacking EGFRvIII expression, compared to those with EGFR amplification and EGFRvIII expression [69]. By studying the results of trials, patterns arise that can lead to more precise pharmacological treatment based on EGFR biomarkers. More development is needed for a method that would allow for a practical identification of EGFR configurations and variants in each patient. These trials’ results and others can be further investigated to select patient populations that are better suited for anti-EGFR treatment.

## 7. Challenges and Potential Strategies

Preclinical and clinical studies have shown that there are limitations in EGFR inhibitor efficacy in glioblastoma treatments. EGFR inhibitor drugs show variable success in effectively penetrating the BBB to target the tumor without reaching toxic levels (Figure 1). There are certain populations of patients with glioblastoma that respond better to EGFR treatment. These drugs also do not show a substantial inhibition of growth pathways alone and must be paired with a synergistic drug to overcome resistance and achieve tumor suppression.

Designing a drug that overcomes the BBB is difficult to accomplish and crucial for any treatment used for glioblastoma. Nanodrugs allow for precise and safe delivery. Their small size and unique properties allow for promising BBB penetration (Figure 1) [70]. A few nanodrugs have been developed for use in patients with glioblastoma. In one, a polymalic acid-based nanoconjugate was attached to drugs targeting protein kinase CK2, EGFRvIII, and wildtype EGFR and was tested in glioblastoma mouse models. Protein kinase CK2 and EGFR act on many of the same pathways. This nanodrug allowed for a synergistic inhibition and effectively decreased oncogenic markers and increased survival in xenogeneic mice, with minimal toxicity [71]. Another nanodrug, poly(amidoamine) dendrimer-based carriers modified with angiopep-2 peptides, binds the low-density lipoprotein receptor-relative protein to achieve BBB penetration. When this delivery system was paired with an EGFR-targeting peptide, high tumor penetrance and increased overall survival were observed in vitro and in vivo [72]. Multifunctional nanopolymers (MNPs) developed based on poly(β-L-malic) acid are another nanodrug suitable for central nervous system treatment. MNPs have the capacity to cross the BBB and are covalently attached to antisense oligonucleotides to target glioblastoma cells by suppressing EGFR/EGFRvIII and c-Myc nuclear transcription factors. When combined with a polymer attached to anti-programmed cell death protein 1 (PD-1) antibody, this drug blocked growth and increased survival in tumor-bearing animals [73]. These nanodrugs display safety and efficacy in crossing the BBB and targeting glioblastoma cells. The diverse classes of nanodrugs being developed show promise to improve tumor targeting and reduce chemotherapy toxicities. Additionally, many of these nanodrugs allow for the combined delivery of synergistic therapeutics. Further trials are needed until nanodrugs become common practice in glioblastoma treatment. Another way the BBB can be overcome is through convection-enhanced delivery (CED). CED locally delivers micro doses of drugs to the CNS via the placement of catheters. It has been used in various trials with glioblastoma treatment and has been found to be safe and viable. However, it had poor results in randomized phase III trials, demonstrating the need for technical improvements and pairing to a drug that targets glioblastoma well [74,75]. Both nanodrugs and CED need further development in trials before their use becomes normalized as they provide hope for improved therapies and reduced toxicities in the near future.

Not all glioblastoma tumors are EGFR driven and there is much heterogeneity within tumors. The development of a diagnostic tool to identify patient populations that may benefit from EGFR inhibitors would inspire narrowly targeted treatment. Identifying the predominant type of mutation and presence of wildtype receptors in patients can help determine which EGFR therapy is viable. One-way in which EGFR configuration can be determined in patients is by the use of anti-EGFR antibody 806i, which is a radiolabeled version of monoclonal antibody 806. It was found in phase 1 trials that antibody 806i had low uptake in normal tissue, a high tumor penetrance indicating a strong ability to cross the BBB, and low toxicity (Figure 1). The use of this antibody followed by computerized tomography (CT) imaging allowed for real-time visualization of EGFR conformation distribution in glioblastoma [76]. In glioblastoma, antibody 806 targets epitope 806, which is present in both amplified wildtype EGFR and EGFRvIII. Antibody 806i allows clinicians to visualize the prevalent EGFR conformations in each patient and determine which drugs to administer in order to target each variant. Recognizing and identifying EGFR mutants in each patient is a vast improvement in the direction of overcoming glioblastoma heterogeneity and establishing targeted EGFR treatment.

Trials until now have shown that EGFR inhibitors yield the best results when used in combination with different therapies. Most EGFR drugs have been tried with radiation and temozolomide and have shown variable results, mostly unsuccessful [51,58,59,60,63]. Amplified wildtype EGFR and variant EGFR enable irregulated growth through various different growth pathways in glioblastoma. EGFR inhibitors do not allow for the sufficient suppression of these pathways and leave room for resistance mutations to occur. Combining EGFR inhibitors with a drug that has synergistic effects on the same downstream pathways may overcome resistance and improve treatment outcomes. High understanding of the downstream pathways of EGFR in glioblastoma is needed to configure ideal multi-drug therapies targeting them. Some potential synergistic downstream targets include mTOR, PI3k, and HDACs [77]. Until now, preclinical trials and clinical trials have shown success in combining anti-EGFR drugs with HDAC inhibitors, rapamycin, bevacizumab, and irinotecan. Further studies are required to confirm a combination of drugs that is consistently effective [52,53,64,65]. Research of downstream pathways could elucidate a way to safely and synergistically inhibit tumor cells with EGFR inhibitors.

## 8. Future Perspectives and Conclusions

Advances in personalized medicine and precision oncology give rise to new horizons in the treatment of highly heterogenous, unique, and invasive glioblastomas. As the knowledge of glioblastoma heterogeneity and its molecular landscape improves, new opportunities to target these tumors arise. A strong understanding of EGFR mutations along with methods of imaging or detecting mutants will elucidate tumor characteristics that can be targeted by small-molecule inhibitors and antibodies to improve patient outcomes.

To effectively target EGFR in glioblastoma, continued research into its signaling pathways is necessary. Research of the structure and downstream pathways of oncogenic EGFR in glioblastoma will allow for the development of revolutionary drugs that specifically target them. Additionally, it allows for pairing with a drug that will synergistically inhibit tumor growth. As the development of next-generation inhibitors continues, increased understanding of EGFR will create more effective targets for these drugs.

EGFR small-molecule inhibitors and antibodies have shown promise in preclinical studies and progress in clinical trials. They need to be further optimized to overcome treatment challenges in glioblastoma, including the BBB and intrinsic resistance. Patient populations or biomarkers that lead to improved outcomes with EGFR inhibitors are needed to guide treatment options. Overall, EGFR proves to be an inspiring horizon to increase treatment options for patients with glioblastoma; however, to practicalize the use of EGFR inhibitors, more trials are needed to configure and personalize treatment.

## Figures and Tables

**Figure 1 ijms-25-02316-f001:**
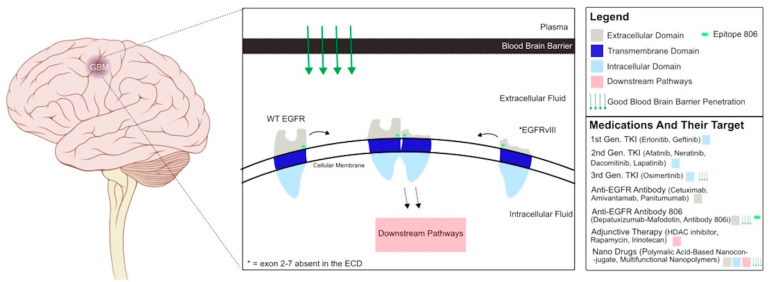
Depicts EGFRvIII and wildtype EGFR dimerization and signaling within a glioblastoma tumor. Tyrosine kinase inhibitors (TKIs) inhibit the intracellular domain of the receptor, while antibodies target the extracellular domain. Adjunctive therapies such as irinotecan target downstream pathways of the EGFR. Finally, nano drugs can be conjugated to drugs previously mentioned to target extracellular and intracellular domains as well as downstream pathways while having strong blood-brain barrier penetration.

**Table 1 ijms-25-02316-t001:** Selected clinical trials of EGFR TKI in Glioblastoma. OS = overall survival, PFS = progression free survival 6 months.

Trial	Regimen	Target Dose	Side Effects	Prospective Trials in Glioblastoma	Summary of Results
A Phase II Study Of Bevacizumab And Erlotinib After Radiation And Temozolomide In MGMT Unmethylated Gbm Patients.	Bevacizumab And Erlotinib	Bevacizumab (intramuscular) at 10 mg/kg every 2 weeks Erlotinib (oral) 150 mg/day. Temozolomide 75 mg/m^2^/day	Lymphopenia, rash, hypertension, fatigue	No active clinical trials	No improvement in OS
Phase II trial of bevacizumab and erlotinib in patients with recurrent malignant glioma	Bevacizumab And Erlotinib	Bevacizumab (intravenously) (10 mg/kg) every 2 weeks Erlotinib (oral) 200 mg/day	Rash, mucositis, diarrhea, and fatigue	No active clinical trials	Drug combination showed activity but resulted in no improved PFS or radiaographic response
A Phase II And Pharmacokinetic Study Of Erlotinib And Sorafenib For Patients With Progressive Or Recurrent Glioblastoma Multiforme.	Erlotinib And Sorafenib	Erlotinib (oral) 150 mg/day sorafenib (oral) 400 mg twice daily for 28 days	Fatigue, diarrhea, hypophosphatemia, acneiform rash	1 active trial for sorafenib	Erlotinib did not improve OS, even when combined with temozolomide, radiation, and other synergistic drugs
Analysis Of Glioblastoma Tissue After Preoperative Treatment With The EGFR Tyrosine Kinase Inhibitor Gefitinib–A Phase II Trial.	Gefitinib	500 mg gefitinib (oral) 5 days prior to surgery	Not reported	no active clinical trials	Gefitinib alone was not enough to stop tumor growth signaling
Phase II Trial Of Dacomitinib In Recurrent Glioblastome Patients With EGFR Amplification.	Dacomitinib	Dacomitinib (oral) 45 mg/day	Rash, diarrhea, asthenia, nausuea/vomitting	no active clinical trials	No improvement in OS
Phase II Trial Of GC1118, A Novel Anti-EGFR Antibody, For Recurrent Glioblastoma Patients With EGFR Amplification.	GC1118	4 mg/kg weekly on Days 1, 8, 15, and 22 of a 28-day cycle x6	Rash, acneiform, mucositis, diarrhea	no active clinical trials	No improvement in progression free survival (PFS)
Depatuxizumab Mafodotin In EGFR-Amplified Newly Diagnosed Glioblastoma: A Phase III Randomized Clinical Trial.	Depatuxizumab-Mafodotin	Dosed at 2.0 mg/kg during radiation therapy, then 1.25 mg/kg thereafter on days 1 and 15/28	Corneal epitheliopathy	no active clinical trials	No improvement in OS, PFS improved in all patients and more so in patients with the EGFRvIII variant
Cetuximab, Bevacizumab, And Irinotecan For Patients With Primary Glioblastoma And Progression After Radiation Therapy And Temozolomide: A Phase II Trial.	Cetuximab, Bevacizumab, And Irinotecan	Patients received bevacizumab (10 mg/kg) (intravenous) Irinotecan (125 mg/m^2^) (Intravenous) and Cetuximab (400 mg/m^2^ as the loading dose on day 1 followed by 250 mg/m^2^ weekly) every 2 weeks for up to 6 months	Nausua, vomitting, diarrhea, stomatisits	4 active trials for cetuximab	Improved median OS, with skin toxicities

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
