# Peer review of "Epidermal Growth Factor Receptor Inhibitors in Glioblastoma: Current Status and Future Possibilities"

_ijms, 2024, doi:10.3390/ijms25042316_

Round 1
Reviewer 1 Report
Comments and Suggestions for Authors
The review from Ezzati et al summarizes the current progression and setbacks of the use of EGFRi in the treatment of glioblastoma.
The review gives a compact description of EGFR and the action of its inhibitors, as well as their use in different types of tumors and the challenges that those molecules face for glioblastoma, supplemented by an informative table of EGFRi clinical trials and their outcome.
The quality of the review is high, and it is well written so it is reader-friendly, while at the same time, it shares all the needed information and messages.
My only major comment is the complete absence of the reference to HER2 contribution, as it is one common suspect for the creation of EGFR heterodimers, while in high concentration it can generate homodimers and activate similar cascades as EGFR. Could this be a reason that in some patients EGFRi present a lower effect than in others? If the authors find this comment relevant, probably they could make a small discussion/ reference about HER2.
Also, take into consideration that abstract should be max 200 words.
Some other minor comments:
- Introduction of the abbreviation GBM is missing. Probably line 39 is a good fit for that.
- Line 59: have been improved
- Explanation of the abbreviation TKIs is only mentioned in the legend of the figure, while it is mentioned earlier in the text
- Line 180: two dots around ref 45
- In vivo and in vitro could be italics
- Sentence in lines 289-219 needs to be rephrased (This is because…)
- Line 220: is variant proper there?
- Line 243: showed that the drug..
- Line 251: double use of word target
- Line 257: ‘and more so in’ doesn’t make sense
- Sentence in lines 262-263 needs to be rephrased (However, …)
- Lines 264-265: ‘achieve BBB penetration’. Is this correct or the increased dose is to achieve a therapeutic dose in the brain/ tumor due to the presence of BBB?
- Lines 267-268 need to be rephrased
- Line 276 needs to be rephrased (Analyzing the…)
- Line 292: And finally needs to be replaced
- Line 306: double in
- Line 370: ‘select patient populations’ needs to be rephrased
Comments on the Quality of English LanguageThe quality of english is fine. Some minor changes for a better understanding are needed and mentioned.
Author Response
To the Editorial team,
We would like to thank the reviewers for their constructive feedback. Implementing the recommended edits has greatly improved the manuscript. Please find below our response to comments from reviewers:
Reviewer 1:
Comments to the Authors: “The review from Ezzati et al summarizes the current progression and setbacks of the use of EGFRi in the treatment of glioblastoma.
The review gives a compact description of EGFR and the action of its inhibitors, as well as their use in different types of tumors and the challenges that those molecules face for glioblastoma, supplemented by an informative table of EGFRi clinical trials and their outcome.
The quality of the review is high, and it is well written so it is reader-friendly, while at the same time, it shares all the needed information and messages.
My only major comment is the complete absence of the reference to HER2 contribution, as it is one common suspect for the creation of EGFR heterodimers, while in high concentration it can generate homodimers and activate similar cascades as EGFR. Could this be a reason that in some patients EGFRi present a lower effect than in others? If the authors find this comment relevant, probably they could make a small discussion/ reference about HER2.”
Author response and action taken:
We thank the reviewer for their valuable feedback regarding the role of HER2 in glioblastoma heterogeneity. In response to this suggestion, we have revised the manuscript to include additional details on of how HER2 contributes to this heterogeneity. Specifically, we have incorporated points discussing how EGFR can dimerize with HER2 receptors and how their upregulation can lead to resistance.
We acknowledge that this aspect was overlooked in previous discussions of heterogeneity, and we appreciate the reviewer’s insight. Having said that, we respectfully made the decision not to include a full section on HER2 due to concerns about the scope of the paper. While HER2 does play a role in glioblastoma resistance, the primary focus of our study is on EGFR mutant-driven glioblastoma and the drugs designed to target these specific mutations. Introducing discussions on targeting HER2 may potentially introduce reader’s confusion, especially considering our findings that suppression of EGFR mutants alone is sufficient for tumor regression.
We thank the reviewer for the comment, the manuscript was changed to reflect this edit.
Page and line number: Page 2, Lines 125, 135-136, page 3, line 195
Comments to the Authors:
“Also, take into consideration that abstract should be max 200 words.”
Author response and action taken:
We thank the reviewer for the comment, the abstract was edited accordingly.
Page and line number: page 1, line 11-26
Comments to the Authors:
““Introduction of the abbreviation GBM is missing. Probably line 39 is a good fit for that.”
Author response and action taken: We thank the reviewer for the comment, the manuscript was changed to reflect this edit.
GBM was removed as an abbreviation and all were changed to “glioblastoma”
Comments to the Authors:
Line 59: have been improved.
We thank the reviewer for the comment, the manuscript was changed to reflect this edit.
Page and line number: page 2, line 100
Comments to the Authors:
“Explanation of the abbreviation TKIs is only mentioned in the legend of the figure, while it is mentioned earlier in the text.”
We thank the reviewer for the comment, the manuscript was changed to reflect this edit.
Page and line number: page 3, line 190
Comments to the Authors:
“Line 180: two dots around ref 45.”
The manuscript was changed to reflect this edit.
Page and line number: page 4, line 235
Comments to the Authors:
“In vivo and in vitro could be italics.”
The manuscript was changed to reflect this edit.
Page and line number: in vivo, and in vitro italicized throughout the manuscript.
Comments to the Authors:
“Sentence in lines 289-219 needs to be rephrased (This is because…).”
The manuscript was changed to reflect this edit.
Page and line number: Page 6, line 279
Comments to the Authors:
“Line 220: is variant proper there?”
We thank the reviewer for the comment, the manuscript was changed to reflect this edit.
Page and line number: page 6, line 280
Comments to the Authors:
“Line 243: showed that the drug..”
We thank the reviewer for the comment, the manuscript was changed to reflect this edit.
Page and line number: page 6, line 302
Comments to the Authors:
“Line 251: double use of word target.”
The manuscript was changed to reflect this edit.
Page and line number: page 6, line 309-311
Comments to the Authors:
“Line 257: ‘and more so in’ doesn’t make sense.”
The manuscript was changed to reflect this edit.
Page and line number: page 6, line 316
Comments to the Authors:
“Lines 264-265: ‘achieve BBB penetration’. Is this correct or the increased dose is to achieve a therapeutic dose in the brain/ tumor due to the presence of BBB?”
We thank the reviewer for bringing this up, the manuscript was changed to reflect this edit.
Page and line number: page 6, line 321- 324
Comments to the Authors:
“Lines 267-268 need to be rephrased.”
The manuscript was changed to reflect this edit.
Page and line number: page 7, line 341-342
Comments to the Authors:
“Line 276 needs to be rephrased (Analyzing the…).”
The manuscript was changed to reflect this edit.
Page and line number: page 7, line 350-351
Comments to the Authors:
“Line 292: And finally needs to be replaced.”
The manuscript was changed to reflect this edit.
Page and line number: page 7, line 366
Comments to the Authors:
“Line 306: double in.”
The manuscript was changed to reflect this edit.
Page and line number: page 8, line 394
Comments to the Authors:
“Line 370: ‘select patient populations’ needs to be rephrased.”
The manuscript was changed to reflect this edit.
Page and line number: page 9, line 460-461
Reviewer 2 Report
Comments and Suggestions for Authors
In this review article, the authors focus on EFGR inhibitors that have been tested for the treatment of glioblastoma and present some of the recent approaches for a more effective EGFR targeting. Initially, the authors describe the different EGFR variants in GBM as well as the most common EGFR inhibitors, focusing exclusively on kinase inhibitors and antibodies. Furthermore, the authors cite several preclinical and clinical studies that use kinase inhibitors or antibodies for EGFR targeting and GBM treatment. Lastly, they are analysing some of the barriers for EGFR inhibition and how they can be overcome through the development of nanodrugs, use of convection-enhanced delivery and combination with other agents.
Overall, the work is consistent with the aims and the scope of the journal. It’s well-structured and its subject is of great importance in the field of glioblastoma. However, I believe that it’s missing crucial aspects concerning the EGFR inhibition. Not only receptor kinase inhibitors and antibodies targeting EGFR have been developed and tested for GBM treatment. There are novel approaches like peptide vaccines (including rindopepimut) and CAR T cell therapies dominating the latest studies. All these approaches have already been described thoroughly in recent reviews https://www.mdpi.com/1422-0067/24/13/11110, https://academic.oup.com/neuro-oncology/article/24/12/2035/6705404, https://www.nature.com/articles/s41388-017-0045-7. Moreover, the description of the clinical data only includes a small number of selected trials.
For the above reasons, I believe that this work in its current form is not suitable for publication as it lacks significant information on the current advancements in EGFR inhibition. Also, there are similar reviews in the literature covering the same subject and thus this work cannot be considered novel.
Author Response
To the Editorial team,
We would like to thank the reviewers for their constructive feedback. Implementing the recommended edits has greatly improved the manuscript. Please find below our response to comments from reviewers:
Reviewer 2:
Comments to the Authors: “In this review article, the authors focus on EFGR inhibitors that have been tested for the treatment of glioblastoma and present some of the recent approaches for a more effective EGFR targeting. Initially, the authors describe the different EGFR variants in GBM as well as the most common EGFR inhibitors, focusing exclusively on kinase inhibitors and antibodies. Furthermore, the authors cite several preclinical and clinical studies that use kinase inhibitors or antibodies for EGFR targeting and GBM treatment. Lastly, they are analysing some of the barriers for EGFR inhibition and how they can be overcome through the development of nanodrugs, use of convection-enhanced delivery and combination with other agents.
Overall, the work is consistent with the aims and the scope of the journal. It’s well-structured and its subject is of great importance in the field of glioblastoma. However, I believe that it’s missing crucial aspects concerning the EGFR inhibition. Not only receptor kinase inhibitors and antibodies targeting EGFR have been developed and tested for GBM treatment. There are novel approaches like peptide vaccines (including rindopepimut) and CAR T cell therapies dominating the latest studies. All these approaches have already been described thoroughly in recent reviews https://www.mdpi.com/1422-0067/24/13/11110, https://academic.oup.com/neuro-oncology/article/24/12/2035/6705404, https://www.nature.com/articles/s41388-017-0045-7. Moreover, the description of the clinical data only includes a small number of selected trials.
For the above reasons, I believe that this work in its current form is not suitable for publication as it lacks significant information on the current advancements in EGFR inhibition. Also, there are similar reviews in the literature covering the same subject and thus this work cannot be considered novel.”
Author response and action taken:
We thank the reviewer for their valuable feedback regarding crucial aspects of EGFR inhibition. We acknowledge that our manuscript primarily focuses on receptor kinase inhibitors and antibodies targeting EGFR, thereby excluding novel approaches such as peptide vaccines (including rindopepimut) and CAR T cell therapies dominating recent studies.
Page and line number: Page 2, Lines 117
In response to reviewer’s comment, we acknowledge the omission of peptide vaccines and CAR T cell therapies from our paper. We have revised the manuscript to reflect a narrower scope, focusing on chemotherapeutic drugs, which aligns with our original intent. While we understand that EGFR drugs encompass a broad range of targets, our paper aims to provide a summary of available treatments and insights for enhancing their efficacy in glioblastoma treatment.
While our paper may not introduce entirely novel information, it offers a concise and accessible summary of EGFR inhibitors in glioblastoma, emphasizing practical insights derived from previous trials and reviews rather than delving extensively into biochemical details. Additionally, we believe that Figure 1 provides a novel contribution by succinctly summarizing key aspects of EGFR dimerization, mutations, downstream pathways, and drug targets, facilitating easy comprehension for readers.
Regarding the selection of trials, we opted to include a small number of trials that presented groundbreaking results for each drug. These trials were chosen either as the latest or most recent, effectively highlighting the limitations and advancements of EGFR inhibitors in glioblastoma treatment.
Round 2
Reviewer 2 Report
Comments and Suggestions for Authors
I agree with the authors that their work provides insights for the improvement of TKI efficacy. However, they should make it clear in both abstract and introduction that their work focuses exclusively on small molecule inhibitors and antibodies. In this way, it won’t be disorientating to the readers.
Author Response
To the editorial team,
We would like to thank the reviewers for their thoughtful feedback. The suggested edits have significantly enhanced the manuscript. Enclosed below is our response to the reviewers' comment:
Reviewer 2:
Comments to the authors: “I agree with the authors that their work provides insights for the improvement of TKI efficacy. However, they should make it clear in both abstract and introduction that their work focuses exclusively on small molecule inhibitors and antibodies. In this way, it won’t be disorientating to the readers.”
Author response and action taken: We sincerely appreciate the reviewer's valuable feedback. Ensuring clarity regarding the focus of our work is essential, and we are grateful for the suggestion. We revised the abstract, introduction, and conclusion to explicitly specify our exclusive focus on small molecule inhibitors and antibodies. Thank you once again for bringing this to our attention.
Page and line number: Page 1, line 14. Page 2, line 123-124. Page 9, line 463-464, line 471.